# Endoplasmic Reticulum-Targeted Phototherapy Remodels the Tumor Immunopeptidome to Enhance Immunogenic Cell Death and Adaptive Anti-Tumor Immunity

**DOI:** 10.3390/ph18040491

**Published:** 2025-03-28

**Authors:** Weidong Xiao, Mingquan Gao, Banghui Mo, Xie Huang, Zaizhi Du, Shufeng Wang, Jianhong Chen, Shenglin Luo, Haiyan Xing

**Affiliations:** 1Department of Pharmacy, Daping Hospital, Third Military Medical University (Army Medical University), Chongqing 400042, China; 2State Key Laboratory of Trauma and Chemical Poisoning, Institute of Combined Injury, Chongqing Engineering Research Center for Nanomedicine, College of Preventive Medicine, Third Military Medical University (Army Medical University), Chongqing 400038, China; 3Department of Immunology, College of Basic Medicine, Third Military Medical University (Army Medical University), Chongqing 400038, China

**Keywords:** ER-targeted phototherapy, immunogenic cell death, tumor immunopeptidome remodeling, endoplasmic reticulum stress, immunogenic peptides, adaptive anti-tumor immunity

## Abstract

**Background:** Endoplasmic reticulum (ER)-targeted phototherapy has emerged as a promising approach to amplify ER stress, induce immunogenic cell death (ICD), and enhance anti-tumor immunity. However, its impact on the antigenicity of dying tumor cells remains poorly understood. **Methods:** Laser activation of the ER-targeted photosensitizer ER-Cy-*po*NO_2_ was performed to investigate its effects on tumor cell antigenicity. Transcriptomic analysis was carried out to assess gene expression changes. Immunopeptidomics profiling was used to identify high-affinity major histocompatibility complex class I (MHC-I) ligands. In vitro functional studies were conducted to evaluate dendritic cell maturation and T lymphocyte activation, while in vivo experiments were performed by combining the identified peptide with poly IC to evaluate anti-tumor immunity. **Results:** Laser activation of ER-Cy-*po*NO_2_ significantly remodeled the antigenic landscape of 4T-1 tumor cells, enhancing their immunogenicity. Transcriptomic analysis revealed upregulation of antigen processing and presentation pathways. Immunopeptidomics profiling identified multiple high-affinity MHC-I ligands, with IF4G3_986–994_ (QGPKTIEQI) showing exceptional immunogenicity. In vitro, IF4G3_986–994_ promoted dendritic cell maturation and enhanced T lymphocytes activation. In vivo, the combination of IF4G3_986–994_ with poly IC elicited robust anti-tumor immunity, characterized by increased CD8^+^ T lymphocytes infiltration, reduced regulatory T cells (Tregs) in the tumor microenvironment, elevated systemic Interferon-gamma (IFN-γ) levels, and significant tumor growth inhibition without systemic toxicity. **Conclusions:** These findings establish a mechanistic link between ER stress-driven ICD, immunopeptidome remodeling, and adaptive immune activation, highlighting the potential of ER-targeted phototherapy as a platform for identifying immunogenic peptides and advancing peptide-based cancer vaccines.

## 1. Introduction

Phototherapy, encompassing photodynamic therapy (PDT) and photothermal therapy (PTT), offers spatiotemporal selectivity, negligible drug resistance, and no cumulative toxicity, making it a promising noninvasive cancer treatment [1,2]. In PDT, a photosensitizer generates reactive oxygen species (ROS) from absorbed light, leading to oxidative damage and subsequent cancer cell death, while PTT converts near-infrared light into heat to damage malignant cells under light irradiation [3,4]. Recent studies have emphasized that phototherapy not only directly eliminates tumor cells but also induces immunogenic cell death (ICD), during which dying tumor cells spatiotemporally release damage-associated molecular patterns (DAMPs), such as calreticulin (CRT), high mobility group box 1 protein (HMGB1), adenosine triphosphate (ATP), and heat shock proteins (HSPs), thereby activating the immune system [5,6,7]. Additionally, the immunogenicity of phototherapy-induced dying cancer cells plays a critical role in mediating anti-tumor immune responses [8]. Nevertheless, several critical challenges, including insufficient tumor selectivity, limited tissue penetration, and the intricate tumor microenvironment, substantially restrict the clinical translation of phototherapy, compromising its therapeutic outcomes, especially its capacity to elicit effective anti-tumor immune responses [9,10,11]. Consequently, developing innovative strategies to overcome these fundamental barriers could enhance the efficacy of phototherapy in inducing ICD, thereby activating specific immunogenicity and improving therapeutic effects [11,12,13].

Recently, an increasing number of studies have shown that the selective delivery of photosensitizer to specific subcellular organelles can significantly influence the phototherapy-induced immunotherapeutic effects [14,15]. The endoplasmic reticulum (ER), as the largest organelle in eukaryotic cells, plays a central role in the maintenance of intracellular signal transduction, calcium homeostasis, and the synthesis and processing of proteins [16,17,18]. As reported, ER dysfunction, exacerbated by ROS-induced protein misfolding, triggers the unfolded protein response (UPR) and subsequent ER stress [19,20,21]. Severe and prolonged ER stress can overwhelm the adaptive capacity of tumor cells, inducing ICD, which subsequently promotes anti-tumor immune responses [22,23]. Therefore, the ICD-associated immunogenicity of tumor cells is largely determined by the duration and intensity of the ER stress [24]. Both our group and others have demonstrated that ER-targeted phototherapy significantly amplifies ER stress, markedly enhances ICD, and promotes the release of DMAPs in tumor cells, thereby eliciting robust anti-tumor immune responses and effectively suppressing the growth of both primary and distant metastatic tumors [25,26]. However, the precise mechanisms by which ER-targeted phototherapy enhance ICD induction and the immunogenicity of dying tumor cells remain to be elucidated [27,28].

Recent studies have shown that the immunogenicity of phototherapy- induced ICD is primarily determined by adjuvanticity, driven by the release of DAMPs, and antigenicity, which is mainly associated with tumor antigens derived from dying cancer cells [29,30]. The adjuvanticity of hallmark DAMPs induced by phototherapy in promoting anti-tumoral immune responses has been extensively investigated, with their release mechanisms thoroughly characterized and their roles in enhancing dendritic cells’ (DCs) maturation and activation being well-established [31,32,33]. However, the impact and mechanisms of phototherapy on the antigenicity of dying tumor cells have been poorly investigated [30,34].

Building on our previous findings that the ER-targeted photosensitizer ER-Cy-*po*NO_2_ amplifies ER stress and enhances the adjuvanticity of ICD tumor cells through the release of DAMPs [26], this study focuses on the underexplored aspect of antigenicity. Specifically, we investigated how ER-Cy-*po*NO_2_-mediated phototherapy reshaped the major histocompatibility complex class I (MHC-I)-associated immunopeptidome (MIP) of dying cancer cells, thereby enhancing ICD and activating robust anti-tumor immune responses (Figure 1). Our findings highlight the dual role of ER-targeted phototherapy in modulating both adjuvanticity and antigenicity, providing a novel approach to suppress tumor growth and metastasis.

## 2. Results

### 2.1. ER-Targeted Phototherapy Enhanced the Immunogenicity of 4T-1 Tumor Cells Undergoing ICD

As we have previously reported, the ER-targeted photosensitizer ER-Cy-*po*NO_2_ can mediate a synergistic PDT/PTT effect that enhances tumor cell ER stress and activates in situ ICD, thereby eliciting a robust systemic anti-tumor immune response [26]. Furthermore, existing evidence suggests that the immunogenicity of cells undergoing stress-driven ICD is a key determinant in activating the adaptive immune response [35]. Therefore, we firstly analyzed the immunogenicity of 4T-1 cells treated with ER-Cy-*po*NO_2_ plus laser irradiation through the IFN-γ Enzyme-linked immunospot (ELISPOT) assay, which is the most widely used method for evaluating immunogenicity in vitro [36,37]. As illustrated in Figure 2a,b, the ER-Cy-*po*NO_2_ + Laser group showed a significantly increased number of IFN-γ spots (more than 500), even higher than that in the positive control group (phorbol myristate acetate (PMA) and ionomycin). This indicated that 4T-1 cells subjected to ER-Cy-*po*NO_2_-mediated phototherapy exhibited strong immunogenicity. Inspired by the robust immunogenic profile of the ER-targeted phototherapeutic 4T-1 cells, we further explored their potential as candidate vaccines for anti-tumor immunization according to previous publications (Figure 2c) [33,38]. Compared with the PBS group, immunization with 4T-1 cells undergoing ICD (ICD 4T-1 cells group) significantly activated mice splenocytes, resulting in ELISPOT spots with significant differences (Figure 2d,e). Tumor growth was significantly inhibited in the ICD 4T-1 cells group compared to that in the PBS group, while there was no significant difference in body weight (Figure 2f and Appendix A).

DCs acting as ’initiators’ in the immune response can more efficiently capture, process, and present tumor antigens to CD8^+^ T lymphocytes in the tumor microenvironment, facilitating their activation and proliferation when DCs are mature [39,40]. Thus, we further investigated the maturation of DCs in the lymph nodes on the injection side where 4T-1 cells undergoing ICD were injected. The results indicated that the proportion of mature DCs (CD80^+^CD86^+^) in the lymph nodes of the ICD 4T-1 cells group increased to around 15%, compared to only 5% in the PBS group (Figure 2g,h and Appendix A). Next, we examined the activation of CD8^+^ T lymphocytes. As shown in Figure 2i,j and Appendix A, the proportion of CD3^+^CD8^+^ T lymphocytes in the spleens of the ICD 4T-1 cells group was significantly higher than that in the PBS group, reaching 40% compared to 30%. The proportion of CD8^+^IFN-γ^+^ cells in the ICD 4T-1 cells group significantly rose to around 12%, compared to only about 3% for IFN-γ-producing CD8^+^ T lymphocytes in the PBS group (Figure 2k,l). In addition, we analyzed the infiltration of CD8^+^ T lymphocytes into tumor tissues. The results showed that the proportion of CD3^+^CD8^+^ T lymphocytes in the PBS group tumors was about 20%, whereas it significantly increased to approximately 35% in the ICD 4T-1 cells group (Figure 2m,n). Finally, we investigated the proportion of regulatory T cells (Tregs), which are crucial components of the immunosuppressive tumor microenvironment [41,42]. We found that the proportion of CD4^+^CD25^+^FoxP3^+^ Tregs in the tumor tissues was remarkably decreased in the ICD 4T-1 cells group compared to that in the PBS group (Figure 2o,p). Collectively, the above results demonstrate that the 4T-1 cells undergoing ICD exhibit robust immunogenicity, which enables them to enhance the maturation of DCs, promote the activation and proliferation of CD8^+^ T lymphocytes, and suppress Tregs.

### 2.2. ICD Tumor Cells Induced by ER-Targeted Phototherapy Triggered a Robust Anti-Tumor Immunity in 4T-1-Bearing Mice

To further evaluate the potential of 4T-1 cells undergoing ICD to activate anti-tumor immune responses, we conducted a therapeutic immunization experiment in tumor-bearing mice (Figure 3a). The tumor growth curves and excised tumor images showed that the ICD 4T-1 cells group exhibited significantly inhibited tumor progression compared with the PBS group, while the treatment with 4T-1 cells undergoing ICD showed no significant impact on body weight compared to PBS treatment (Figure 3b,c and S1b, Appendix A). Further analysis of the lymph nodes revealed that the ICD 4T-1 cells group exhibited an enhanced frequency of mature DCs (CD80^+^CD86^+^) in both the primary and distant lymph nodes compared with the PBS group (Figure 3d–g). Moreover, the flow cytometry assessment of the tumor microenvironment showed that the ICD 4T-1 cells group had an increased infiltration of CD3^+^CD8^+^ T lymphocytes and a decreased abundance of immunosuppressive Tregs within the 4T-1 tumors compared to the PBS group (Figure 3h–k). In addition, the serum IFN-γ levels in the ICD 4T-1 cells group further corroborated the induction of a robust anti-tumor compared with the PBS group (Figure 3l). Taken together, these results are in strong agreement with the trends observed in the previous tumor vaccine experiments, further validating that treatment with 4T-1 cells undergoing ICD, mediated by ER-Cy-*po*NO_2_ phototherapy, allows 4T-1 cells to activate the immune system and suppress tumor growth.

### 2.3. ER-Targeted Phototherapy Upregulated Antigen Processing and Immune Signaling Pathways in 4T-1 Cells

Encouraged by the capability of the ICD 4T-1 cells to activate anti-tumor immune responses, we performed transcriptome sequencing analysis to elucidate the underlying mechanism of ER-Cy-*po*NO_2_-mediated phototherapy (Appendix A). As shown in the volcano plot in Figure 4a, compared with the PBS group, there were 2694 differentially expressed genes (DEGs) in the ER-Cy-*po*NO_2_ + Laser treatment group (1459 upregulation and 1235 downregulation), indicating substantial transcriptional changes induced by the ER-targeted phototherapy. The heatmap depicted the DEGs, further highlighting the distinct gene expression profiles between the PBS and the ER-Cy-*po*NO_2_ + Laser treatment groups (Figure 4b).

To reveal the functional categories of DEGs, we conducted GO and KEGG enrichment analyses. The Gene Ontology (GO) analysis revealed that ER-Cy-*po*NO_2_-mediated phototherapy significantly upregulated the expression of genes associated with multiple biological processes, including the cellular stress response and immune response (Figure 4c). The KEGG enrichment analysis further revealed that ER-Cy-*po*NO_2_ + Laser treatment led to the upregulation of genes enriched in immune-related signaling pathways, including the IL-17 signaling pathway, cytokine–cytokine receptor interaction, TNF signaling pathway, calcium signaling pathway, and antigen processing and presentation. In contrast, several tumor-promoting pathways, such as the mTOR signaling pathway, PI3K-Akt signaling pathway, EGFR tyrosine kinase inhibitor resistance, TGF-β signaling pathway, and Ras signaling pathway, were significantly downregulated following the ER-Cy-*po*NO_2_-mediated phototherapy (Figure 4d).

Interestingly, the Gene Set Enrichment Analysis (GSEA) showed that ER-Cy-*po*NO_2_ + Laser treatment significantly upregulated the expression of genes involved in antigen processing and presentation (NES = 1.7396, FDR q = 0.006), indicating the enhanced activation of the immune system (Figure 4e). Moreover, the expression of multiple key genes involved in antigen processing and presentation, including Hspa1b, Hspa1a, HSpa8, HSp90aa1, Hsp90ab1, H2-M10.1, H2-T23, and Psme1, were significantly upregulated by ER-Cy-*po*NO_2_-mediated phototherapy (Figure 4f). Taken together, these results demonstrate that ER-Cy-*po*NO_2_-mediated phototherapy upregulates the expression of genes involved in antigen processing and presentation, cytokine signaling, and immune effector functions, potentially driven by ICD in 4T-1 cells, which subsequently triggers a robust anti-tumor immune response.

### 2.4. ER-Targeted Phototherapy Remodeled Immunopeptidome of 4T-1 Cells, Generating High-Affinity MHC-I Ligands

Considering that the immunopeptide repertoire presented on the surface of tumor cells by MHC-I was a critical determinant of anti-tumor immune responses, we further performed a comprehensive immunopeptidomics analysis to characterize the changes in immunopeptidome on the surface of 4T-1 cells between the PBS and the ER-Cy-*po*NO_2_ + Laser treatment groups (Figure 5a). A total of 6671 peptide segments were identified, with the length distribution of the MHC-I peptides plotted for each group (Figure 5b and Appendix A). A Venn diagram revealed that 720 peptide segments were unique to the PBS group, 2694 were unique to the ER-Cy-*po*NO_2_ + Laser group, and 3257 were shared between the two groups, demonstrating that phototherapy substantially altered the peptides profile of the 4T-1 tumor cells (Figure 5c). Among the shared peptides, 307 were significantly differentially expressed, with 127 upregulated and 180 downregulated following the ER-Cy-*po*NO_2_ + Laser treatment (Figure 5d and Appendix A). The affinities of significantly upregulated peptide segments to MHC-I, including H2-Kd, H2-Dd, and H2-Ld, were predicted using the NetMHCpan tool (ver. 4.1), with strong binders (SBs) defined as a binding affinity score % rank < 0.5 and weak binders (WBs) as % rank of 0.5–2. Among the upregulated peptides, SBs accounted for 71.5% (H2-Dd), 39.2% (H2-Kd), and 31.5% (H2-Ld), respectively (Figure 5e). Furthermore, the immunogenicity rank of the identified peptides was evaluated and ranked using the DeepImmuTM platform, as described in the previous publication [43], with the results shown in Appendix A Sheet 1. The analysis of the corresponding sequence motifs revealed distinct patterns for each haplotype, suggesting that phototherapy-induced changes in the immunopeptidome involved the selective presentation of specific peptide sequences on MHC-I (Figure 5f). Collectively, these findings indicate that ER-Cy-*po*NO_2_-mediated phototherapy induced substantial changes in the immunopeptidome of 4T-1 tumor cells, generating multiple upregulated immunopeptides with high affinity to MHC-I.

### 2.5. The IF4G3_986–994_ Peptide Demonstrated a Pronounced Immunogenicity Advantage over the Other Tested Candidates

Building upon the results of previous transcriptome sequencing and immunopeptidomics analyses, we further systematically screened and evaluated highly immunogenic peptide candidates (Figure 6a and Appendix A). First, 4892 peptides with a length of 8–11 amino acids, the typical range for MHC-I ligands, were selected (Appendix A). Second, 108 peptides that were significantly upregulated were identified (Appendix A). Third, 82 of these upregulated peptides were predicted to have strong MHC-I binding affinity (Appendix A). Finally, the top 15 most immunogenic peptides were selected based on their predicted immunogenicity scores, and their sequences, source proteins, and predicted allele specificity were summarized (Figure 6b and Appendix A). The candidate antigen peptides were then synthesized in vitro, and the mass spectrometry (MS) and high-performance liquid chromatography (HPLC) confirmed that the purity of all peptides exceeded 95% (Appendix A). Among them, the 15th peptide, IF4G3_986–994_ (QGPKTIEQI), elicited the strongest IFN-γ ELISPOT response, even surpassing the positive control (PMA plus ionomycin), which is known to strongly activate T lymphocytes (Figure 6c–e). This suggests that the IF4G3_986–994_ peptide may be the most potent stimulator of mouse immune activation among the tested candidates. To further validate the immunogenicity of the candidate peptides, they were used to stimulate the maturation of mouse BMDCs in vitro (Figure 6f and Appendix A). The maturation of BMDCs was assessed by the upregulation of surface markers, such as CD80 and CD86. Consistent with the IFN-γ ELISPOT results, the 15th peptide, IF4G3_986–994_, showed the strongest ability to promote BMDC maturation among the tested peptides (Figure 6f). Taken together, the comprehensive analysis of IFN-γ ELISPOT and BMDC maturation assays demonstrates that the IF4G3_986–994_ peptide exhibits the most potent immunogenicity and ability to activate immune responses among the 15 candidate peptides evaluated.

### 2.6. The Highly Immunogenic Peptide IF4G3_986–994_ Could Effectively Activate the Anti-Tumor Immune Response and Subsequently Inhibit Tumor Growth in Mice

At last, we further evaluated the immunogenic potential of the immunopeptide IF4G3_986–994_ and its ability to activate anti-tumor immune responses through in vivo tumor vaccine experiments (Figure 7a). Building on the strong immunogenicity of IF4G3_986–994_ demonstrated in vitro, peptide-based vaccines often benefit from the addition of adjuvants to further enhance their immunogenic potential in vivo [44]. Therefore, polyinosinic-polycytidylic acid (poly IC), a synthetic double-stranded RNA analog and agonist for TLR3 and RIG-I, was used as a vaccine adjuvant to amplify both innate and adaptive immune responses [45,46].

In comparison to the PBS and the poly IC alone group, the poly IC + IF4G3_986–994_ group showed significantly increased splenocyte activation, as evidenced by the elevated IFN-γ ELISPOT spots (Figure 7b,c). Additionally, the poly IC + IF4G3_986–994_ group exhibited significantly slower tumor growth, with no significant differences in body weight or blood routine among the groups (Figure 7d, Appendix A). The tumor sizes and weights in the poly IC + IF4G3_986–994_ group were markedly reduced compared to the PBS and poly IC groups (Figure 7e,f). The proportion of mature CD80^+^CD86^+^ DCs in the lymph nodes and CD8^+^ T lymphocytes in the spleen was significantly higher in the poly IC + IF4G3_986–994_ group than that in the PBS group (Figure 7g–j). Notably, the percentage of IFN-γ^+^CD8^+^ T lymphocytes in the spleens reached nearly 30% in the poly IC + IF4G3_986–994_ group, compared to 5% in the PBS group and 15% in the poly IC group (Figure 7k,l). Furthermore, the proportion of CD4^+^CD25^+^FoxP3^+^ Tregs in tumors decreased by approximately 10% in the poly IC + IF4G3_986–994_ group, indicating a reduction in the immunosuppressive tumor microenvironment alongside robust immune activation (Figure 7m,n). Serum Enzyme-Linked Immunosorbent (ELISA) analysis further revealed significantly elevated IFN-γ levels in the poly IC + IF4G3_986–994_ group compared to the PBS and poly IC groups (Appendix A). These results demonstrate that the highly immunogenic IF4G3_986–994_ peptide, combined with poly IC, robustly activates anti-tumor immune responses and inhibits tumor growth in mice.

## 3. Discussion

In this study, we demonstrated that ER-targeted phototherapy significantly enhanced the immunogenicity of 4T-1 tumor cells undergoing ICD. These cells effectively activated DC maturation, promoted CD8^+^ T lymphocyte responses, and suppressed Tregs in both prophylactic and therapeutic models, leading to robust anti-tumor immune responses and tumor growth inhibition. Transcriptomic analyses revealed that ER-Cy-*po*NO_2_-mediated phototherapy upregulated genes involved in antigen processing and immune signaling pathways, while downregulating tumor-promoting pathways, providing a mechanistic basis for the enhanced immunogenicity. Furthermore, immunopeptidomics profiling showed that ER-Cy-*po*NO_2_-mediated phototherapy remodeled the immunopeptidome of 4T-1 cells, generating high-affinity MHC-I ligands. Among these, the IF4G3_986–994_ peptide exhibited exceptional immunogenicity, effectively activating DCs and T lymphocytes in vitro and eliciting potent anti-tumor immunity in vivo when combined with poly IC as an adjuvant. This study establishes a mechanistic link between ER stress-driven ICD, immunopeptidome remodeling, and adaptive immune activation, providing important insights into the underlying mechanisms by which ER-targeted phototherapy enhances tumor cell immunogenicity and induces potent anti-tumor immune responses.

ICD is a distinctive form of regulated cell death that triggers antigen-specific immune responses and enhances anti-tumor immunity [47,48]. Phototherapy has emerged as a promising approach for cancer treatment due to its ability to induce ICD in tumor cells [49,50]. However, the efficacy of phototherapy-induced ICD is often constrained by the insufficient immunogenicity of dying tumor cells, which limits its capacity to activate systemic anti-tumor immunity [51,52]. In this study, we demonstrated that ER-targeted phototherapy using ER-Cy-*po*NO_2_ significantly enhanced the immunogenicity of 4T-1 cells undergoing ICD, as evidenced by the increased production of IFN-γ in ELISPOT assays. This enhanced immunogenicity translated into robust anti-tumor responses in vivo, including DC maturation, the increased infiltration of CD8^+^ T lymphocytes, and a reduction in Tregs within the tumor microenvironment. The observed enhancement in immunogenicity can be attributed to two key factors. First, the ER-targeting capability of ER-Cy-*po*NO_2_ amplified ER stress, which is a critical determinant of ICD-associated immunogenicity, as previously reported [53,54]. Second, the synergistic PDT and PTT effects of ER-Cy-*po*NO_2_ further aggravated ER dysfunction and intensified ER stress [26]. Together, these effects induced robust ICD and enhanced the ability of dying tumor cells to act as immunogenic antigens, thereby promoting systemic immune activation. Compared to existing strategies for inducing ICD [55,56], such as chemotherapy or radiotherapy, our findings underscore the unique advantages of ER-targeted phototherapy in enhancing tumor cell immunogenicity and addressing a key limitation in ICD-based therapies. These results highlight the potential of ER-targeted phototherapy as a promising strategy for ICD-based cancer treatment and provide a strong rationale for its further exploration in combination with other immunotherapeutic approaches.

The immunogenicity of ICD cells is determined by three crucial aspects: adjuvanticity, antigenicity, and the tumor immune microenvironment (TME) [35]. Previous studies by our group have demonstrated that ER-Cy-*po*NO_2_-mediated phototherapy enhances the adjuvanticity of ICD cells by inducing ER stress, which amplifies ICD through the promotion of CRT translocation, HMGB1 release, and ATP secretion. Additionally, we have shown that this phototherapy remodels the TME, increasing the infiltration of DCs and CD8^+^ T lymphocytes while significantly reducing immunosuppressive Tregs and M2 macrophages [26]. Building on these findings, the present study focuses on the less-explored but equally critical aspect of antigenicity, which directly determines the ability of the immune system to recognize and attack tumor cells. Our results reveal that ER-Cy-*po*NO_2_-mediated phototherapy significantly enhances antigen processing and presentation pathways and remodels the immunopeptidome, thereby amplifying the antigenicity of ICD cells and contributing to their robust immunogenicity.

The transcriptomic analysis demonstrated that ER-Cy-*po*NO_2_-mediated phototherapy significantly enhanced antigen processing and presentation pathways, a key factor in tumor antigenicity. The upregulation of genes, such as Hspa1a, Hspa1b, Hspa8, Hsp90aa1, Hsp90ab1, H2-M10.1, H2-T23, and Psme1, as revealed by GSEA and KEGG enrichment analyses, highlighted the role of ER stress in driving these changes. Heat shock proteins (HSP70 and HSP90 families), which were significantly upregulated, were known to stabilize antigenic peptides and enhance their cross-presentation by DCs, thereby promoting CD8^+^ T lymphocyte activation [57,58]. This aligned with the established role of ER stress in enhancing tumor antigenicity, but ER-Cy-*po*NO_2_ -mediated phototherapy provided a more targeted mechanism compared to conventional therapies [59]. The upregulation of Psme1, a key immunoproteasome subunit, further supported enhanced antigen processing, as it facilitated the generation of antigenic peptides for MHC-I loading [60]. Combined with the increased expression of MHC-I-associated genes, such as H2-M10.1 and H2-T23, these findings suggested that ER-Cy-*po*NO_2_-mediated phototherapy promoted efficient antigen generation, processing, and presentation, thereby amplifying tumor antigenicity. Additionally, the downregulation of tumor-promoting pathways, including PI3K-Akt, mTOR, and TGF-β signaling, indicated that this phototherapy not only enhanced immune activation but also suppressed immune evasion mechanisms [61,62,63]. This dual modulation of immune activation and tumor suppression likely contributed to the robust anti-tumor immune responses observed in vivo.

In addition to the transcriptomic changes, immunopeptidomics analysis provided direct evidence that ER-Cy-*po*NO_2_-mediated phototherapy significantly remodeled the immunopeptidome of 4T-1 tumor cells undergoing ICD. Immunopeptidomics primarily focuses on the detection and analysis of peptides that bind to major histocompatibility complex (MHC) molecules, including the ligands of both MHC class I and MHC class II. MHC class I molecules primarily present endogenous antigenic peptides, which typically originate from the degradation products of intracellular proteins. These peptides are critical for immune surveillance, allowing the immune system to recognize and eliminate transformed cells [64]. In our study, immunopeptidomics analysis revealed that ER-targeted phototherapy mediated by ER-Cy-*po*NO_2_ significantly remodels the immunopeptidome of 4T-1 cells. Among the 307 differentially expressed shared peptides, the 127 upregulated peptides exhibited a high proportion of strong binding affinities to MHC-I molecules, with H2-Dd (71.5%) being particularly prominent, followed by H2-Kd (39.2%) and H2-Ld (31.5%). The binding affinity of upregulated peptides to MHC-I alleles, like H2-Kd, H2-Dd and H2-Ld, is crucial for their immunogenic potential. High-affinity peptides form stable MHC–peptide complexes, which are essential for engaging CD8^+^ T lymphocytes. These stable complexes promote CD8^+^ T cell activation and IFN-γ production, driving a robust anti-tumor immune response [65]. This finding indicated that ER-Cy-*po*NO_2_-mediated phototherapy not only altered the peptide repertoire but also selectively enhanced the presentation of high-affinity MHC-I ligands, which are critical for effective CD8^+^ T lymphocytes activation [66,67,68]. Furthermore, the distinct sequence motifs identified for each MHC-I haplotype suggested that ER stress induced by ER-Cy-*po*NO_2_-mediated phototherapy drove the selective presentation of specific peptide sequences, reflecting a targeted remodeling of the immunopeptidome [69]. These results highlighted the unique ability of ER-Cy-*po*NO_2_-mediated phototherapy to amplify tumor antigenicity by generating immunogenic peptides with high MHC-I affinity. This mechanism provided a direct link between the upregulation of immunopeptides and the broader immunostimulatory effects observed in the transcriptomic analysis, such as the activation of immune-related pathways. By directly identifying and quantifying MHC-I-bound peptides, this study offered robust evidence of enhanced antigenicity, supporting the potential of ER-Cy-*po*NO_2_-mediated phototherapy as a powerful strategy for improving anti-tumor immune responses in cancer treatment.

Through further systematic screening based on transcriptomic and immunopeptidomics analyses of ICD 4T-1 tumor cells, we identified IF4G3_986–994_ (QGPKTIEQI) as the most immunogenic peptide among the 15 top-ranked candidates. IF4G3_986–994_ elicited the strongest IFN-γ response in ELISPOT assays and significantly promoted BMDC maturation, demonstrating its superior immunogenicity. These findings provided direct evidence that ER-Cy-*po*NO_2_-mediated phototherapy remodeled the immunopeptidome of ICD tumor cells by enriching high-affinity MHC-I ligands, thereby enhancing their antigenicity and immunogenic potential. In vivo, IF4G3_986–994_, combined with poly IC, further validated its immunogenic potential by significantly enhancing DC maturation, increasing CD8^+^ T lymphocyte activation, and reducing Treg populations. This combination robustly inhibited tumor growth without systemic toxicity, as indicated by stable body weight and normal blood parameters. These findings demonstrated that IF4G3_986–994_ not only activated adaptive immune responses but also modulated the tumor microenvironment to favor anti-tumor immunity. This study confirmed the feasibility of using ER-targeted phototherapy to identify high-immunogenicity peptides, such as IF4G3_986–994_, which could serve as potent immune activators. These findings highlighted the potential of ER-targeted phototherapy to induce ICD and serve as a platform for identifying immunogenic peptides, thus advancing peptide-based vaccine strategies. Compared to ICD tumor cell-based vaccines, peptide-based vaccines offered advantages, such as higher specificity, reduced off-target effects, and improved scalability for clinical applications [70,71].

This study had several limitations that warrant further investigation. First, the anti-tumor mechanisms mediated by IF4G3_986–994_, including its activation of specific signaling pathways and interactions with other immune components, remain incompletely elucidated. Second, tetramer assays to detect IF4G3_986–994_-specific CD8^+^ T lymphocytes and the quantification of cytotoxic molecules (e.g., perforin and granzyme) in CD8^+^ T cells were not performed, limiting the confirmation of antigen-specific effector T lymphocytes in BALB/c mice. Finally, the potential synergistic effects among multiple immunopeptides and their contributions to the overall immune response were not explored. Future studies should aim to address these limitations by further elucidating the mechanisms of action of IF4G3_986–994_, optimizing peptide screening and validation methods, and exploring its potential synergistic effects with other immunogenic peptides or immunotherapies. These efforts will be essential to enhance the anti-tumor immune response of IF4G3_986–994_ and advance the clinical translation of peptide-based vaccines.

## 4. Materials and Methods

### 4.1. Reagents

The following reagents were used in this study. Dyes and antibodies: Fixable Viability Dye eFluor^®^ 780 (#565388, BD, Franklin Lakes, NJ, USA), PE-conjugated anti-CD11c (#12-0114-82, eBioscience, San Diego, CA, USA), APC-conjugated anti-CD80 (#17-0801-82, eBioscience, San Diego, CA, USA), PE Cy7-conjugated anti-CD86 (#25-0862-82, eBioscience, San Diego, CA, USA), FITC-conjugated anti-CD3 (#11-0031-82, eBioscience, San Diego, CA, USA), APC-conjugatedanti-CD45 (#17-0451-82, eBioscience, San Diego, CA, USA), PE Cy7-conjugated anti-CD4 (#25-0041-82, eBioscience, San Diego, CA, USA), PerCP-Cy5.5-conjugated anti-CD8 (#45-0081-82, BioLegend, San Diego, CA, USA), PE-conjugated anti-FoxP3 (#320008, BioLegend, San Diego, CA, USA), BV421-conjugated anti-CD25 (#102043, BioLegend, San Diego, CA, USA), and PE-conjugated anti-IFN-γ (#505807, BioLegend, San Diego, CA, USA). Cytokines and kits: Interleukin-2 (IL-2, #212-12, Peprotech, Cranbury, NJ, USA), granulocyte-macrophage colony-stimulating factor (GM-CSF, #315-03, Peprotech, Cranbury, NJ, USA), interleukin-4 (IL-4, #214-14, Peprotech, Cranbury, NJ, USA), Mouse IFN-γ ELISPOT Kit (#2210005, Dakewei Biotech, Shenzhen, China), and Mouse IFN-γ ELISA Kit (SMK2918A, MEIKE, Zhenjiang, Jiangsu, China).

### 4.2. Synthesis of ER-Cy-poNO_2_

ER-Cy-*po*NO_2_ was synthesized following a previously reported three-step protocol in a straightforward manner [26].

### 4.3. Cell Culture and Treatment

The 4T-1 cells were obtained from the Cell Bank of the Type Culture Collection at the Chinese Academy of Sciences (Shanghai, China) and cultured in complete DMEM medium (Gibco, Billings, MT, USA) containing 10% fetal bovine serum (FBS, Hyclone, Logan, UT, USA) and 1% penicillin/streptomycin (Gibco, Billings, MT, USA). Incubation was carried out at 37 °C in an atmosphere with 5% CO_2_.

For ICD induction, 5 × 10^2^ 4T-1 cells were seeded in 6-well plates and left to adhere overnight. The following day, treatment of the 4T-1 cells was carried out as per the assigned ER-Cy-*po*NO_2_ + Laser and Phosphate-Buffered Saline (PBS, Hyclone, Logan, UT, USA) groups. For the PBS group, only an equal volume of sterile PBS solution was added. After the above experimental procedures, the 4T-1 cells were cultured for another 24 h. Then, they were harvested, counted, and utilized for subsequent experiments.

### 4.4. Immunogenicity Evaluation of 4T-1 Cells

The immunogenicity evaluation experiments were divided into 4 groups with the following treatments. Negative control group: cultured in complete DMEM medium. Laser alone group: cultured in complete DMEM medium and exposed to 808 nm NIR laser irradiation (1 W/cm^2^ for 5 min). ER-Cy-*po*NO_2_ alone group: cultured in DMEM containing 4 μM ER-Cy-*po*NO_2_. ER-Cy-*po*NO_2_ + Laser group: incubated with 4 μM ER-Cy-*po*NO_2_ in DMEM for 24 h, followed by exposure to 808 nm NIR laser irradiation (1 W/cm^2^ for 5 min). The immunogenicity of 4T-1 cells was evaluated using IFN-γ ELISPOT assays. Briefly, 1 × 10^4^ 4T-1 cells that had been subjected to different treatments per well were co-cultured with the newly harvested splenocytes of BALB/c mice (2 × 10^5^ cells per well) in 96-well ELISPOT plates that were pre-coated with a capture antibody against IFN-γ. The co-culture was placed in an incubator at 37 °C with 5% CO_2_. After 20 h, IFN-γ-secreting cells were detected with the ELISPOT kit according to the manufacturer’s protocol. Spot counting was carried out using the spot reader system (Ver 1.0, Saizhi, Beijing, China).

### 4.5. Transcriptome Sequencing Analysis

4T-1 cells were divided into a PBS control group and a treatment group (ER-Cy-*po*NO_2_ + Laser). After 24 h of treatment, 4T-1 cells from each group were collected and adjusted to more than 2 × 10^7^ cells. The cell samples were then frozen in liquid nitrogen for 20 min and subsequently stored at −80 °C. Transcriptome sequencing experiments and bioinformatics analyses were performed by Bioprofile Biotechnology Co., Ltd. (Shanghai, China). DEGs were identified with the criteria of Log_2_(Fold Change) > 1 and adjusted *p*-value < 0.05 (calculated using the Benjamini–Hochberg method) [72]. For GO enrichment analysis, the top 20 upregulated pathways were ranked by ascending false discovery rate (FDR) across all categories. For KEGG enrichment analysis, pathways were screened with an FDR threshold of < 0.05, and cancer-related pathways were highlighted [73].

### 4.6. Immunopeptidomics, Peptide MHC-I-Binding Affinity, and Immunogenicity Prediction

4T-1 cells were also assigned to a PBS control group and a treatment group (ER-Cy-*po*NO_2_ + Laser). A total of 2 × 10^7^ 4T-1 cells from each treatment group were harvested and lysed in buffer following the cell collection methods described in the literature [74]. Briefly, the collected cells were resuspended in PBS and lysed with an equal volume of lysis buffer on ice for 60 min, followed by sonication for 5 min. The lysate was centrifuged at 2000× *g* for 20 min at 4 °C to remove debris, and the supernatant was further clarified by centrifugation at 20,000× *g* for 20 min at 4 °C. The resulting supernatant containing mouse MHC–peptide complexes was enriched using the NEO Discovery MHC-I Peptide Enrichment Kit (Baizhen Biotechnology Co., Ltd., Wuhan, China). The isolated MHC-I-peptide complexes were separated using a reverse-phase column to obtain the immunopeptide fractions, which were then analyzed by HPLC (nanoElute, Bruker, Billerica, MA, USA) with tandem mass spectrometric analysis (LC-MS/MS, timsTOF Pro2, Bruker, MA, USA). Briefly, peptide samples were reconstituted in a formic acid–water solution and separated using HPLC system. The HPLC parameters were optimized as follows: column temperature was maintained at 60 °C, and the flow rate was set to 0.3 μL/min. The mobile phases consisted of 0.1% formic acid (A117-50, ThermoFisher, Waltham, MA, USA) in water (Direct-Q 5UV, Millipore, Billerica, MA, USA) (mobile phase A) and 0.1% formic acid in acetonitrile (34851, Sigma–Aldrich, St. Louis, MO, USA) (mobile phase B). The peptide samples underwent separation via gradient elution. Initiating at 96% mobile phase A and 4% mobile phase B, over 48 min, it gradually shifted to 80% mobile phase A and 20% mobile phase B. By 50 min, the gradient was adjusted to 70% mobile phase A and 30% mobile phase B. Finally, by 55 min, it reached 100% mobile phase B, which was maintained until the 60 min run ended. The separated peptides were then analyzed using a high-resolution MS operating in data-dependent acquisition (DDA) mode. The full mass scan range was set to *m/z* 100–1700, with 10 MS/MS scans per cycle and a total cycle time of 2.27 s. The ion intensity threshold for triggering MS/MS was set to 2500. The LC-MS/MS analysis generated raw MS data files (.d), which were then utilized for subsequent data processing and bioinformatics analysis. The raw mass spectrometry data files generated during the LC-MS/MS analysis were processed and analyzed using the PEAKS DeepNovo Peptidome software (Ver 12.0, Bioinformatics Solutions Inc. Wuhan, China). This platform integrates advanced algorithms, including de novo sequencing, database search, and homology search, to comprehensively identify peptides from the acquired data. The SPIDER algorithm within PEAKS was utilized to detect peptide sequence variants and mutations, which were classified as “Homolog” peptides. Peptides that could not be matched to the reference Mus musculus proteome (Mus_musculus.GRCm39.110) were categorized as “DeepNovo” peptides. To ensure high confidence in the peptide identifications, only those with an average local confidence (ALC) score greater than 80% were retained for further analysis. Given that MHC-I-bound peptides are naturally processed without enzymatic digestion, no enzyme specificity was selected during the database search. The precursor mass and fragment ion mass tolerances were set to 20.0 ppm and 0.05 Da, respectively. Methionine oxidation (15.99 Da) was included as a variable modification, and the FDR for the identified peptides was stringently controlled at 1%. The analysis focused on peptides ranging from 8 to 11 amino acids in length, as this is the typical size range for MHC-I-presented peptides.

The identified peptides were evaluated for their potential binding to the mouse MHC-I alleles H2-Kd, H2-Dd, and H2-Ld using the state-of-the-art MhcVizPipe program and the well-established IEDB analysis resource NetMHCpan-4.1 as described in the previous study [75,76]. This comprehensive assessment provided valuable insights into the MHC-I binding capacity of the detected immunopeptides. Furthermore, the DeepImmuTM Neoantigen Discovery platform (Bioinformatics Solutions Inc.) was employed to predict the immunogenicity of the identified immunopeptides according to the previous study [43,77].

### 4.7. Tumor Vaccine Experiments

Six-week-old female BALB/c mice (weighing 16–20 g) were procured from Weitonglihua Experimental Animal Technology Co., Ltd. (Beijing, China). They were kept with unrestricted access to food and water. All animal-related procedures received the green light from the Ethics Committee of the Third Military Medical University (Chongqing, China; approval number: AMUWEC20224667) and were carried out following national guidelines. The mice were randomly divided into 2 groups. The PBS group was administered with 100 μL sterile PBS, while the treatment group (ICD 4T-1 cells group) received 1 × 10^6^ ICD 4T-1 cells suspended in 100 μL of sterile PBS, injected subcutaneously into the right hind flank. Vaccinations were administered every 3 days for a total of 3 doses. One week after the final vaccination, 3 mice from each group were sacrificed, and DCs were isolated from the lymph nodes for further flow cytometry analysis. The rest of the mice in each group were subsequently inoculated with 1 × 10^6^ normal 4T-1 cells suspended in 100 μL of sterile PBS into the left hind flank. The body weight and tumor volume were measured at intervals of 3 days. The volume of the tumor was computed by applying the modified ellipsoid formula, which is defined as Volume (mm^3^) = (length × width^2^)/2. Mice were humanely euthanized when the average tumor volume of PBS group reached 600 mm^3^ [78], and lymphocytes were isolated from both the spleen and tumor tissues for flow cytometry analysis.

### 4.8. Tumor Treatment Experiments

Six-week-old female BALB/c mice were randomly divided into 2 groups. A total of 1 × 10^6^ 4T-1 cells in 100 μL sterile PBS were inoculated into the right hind flank of each mouse to establish the 4T-1 xenograft model. Three days later, the PBS group received 100 μL of sterile PBS, while the treatment group (ICD 4T-1 cells group) received 1 × 10^6^ ICD 4T-1 cells in 100 μL sterile PBS, administered subcutaneously into the left hind flank. Treatment occurred every 3 days, with a total of 3 doses administered. The body weight and tumor volume were measured at intervals of 3 days. Mice were humanely euthanized when the average tumor volume of PBS group reached 600 mm^3^, and the serum were collected for further ELISA analysis. The levels of IFN-γ in the serum were quantified using a Mouse IFN-γ ELISA Kit following the manufacturer’s instructions. DCs and lymphocytes were collected for further flow cytometry analysis, and the tumor tissues were also harvested for weighing and imaging.

### 4.9. In Vivo Immunogenicity Assessment of Candidate Peptides

Six-week-old female BALB/c mice were randomly allocated into 3 groups. The negative control group was administered with 100 μL of sterile PBS. The poly IC control group received 10 μg of poly IC in 100 μL sterile PBS. The treatment group received 10 μg of the candidate peptides IF4G3_986–994_ combined with 10 μg of poly IC in 100 μL sterile PBS. All injections were given subcutaneously at three-day intervals, with a total of 3 doses being administered. Seven days after the final vaccination, spleens from 3 mice in each group were harvested for IFN-γ ELISPOT assay. The mice remaining in each group were then inoculated with 1 × 10^6^ normal 4T-1 cells subcutaneously into the right hind flank. Body weight and tumor volume were measured every 3 days, while routine blood tests were performed weekly. Mice were humanely euthanized when the average tumor volume of PBS and poly IC control group reached 600 mm^3^, and serum samples were collected for subsequent ELISA analysis. DCs and lymphocytes were collected for subsequent flow cytometry analysis, and the tumor tissues were also harvested for weighing and imaging.

### 4.10. Flow Cytometry Analysis

To analyze the maturation of DCs, cells isolated from the lymph nodes were stained with Fixable Viability Dye eFluor^®^ 780, PE-conjugated anti-CD11c, APC-conjugated anti-CD80, and PE Cy7-conjugated anti-CD86. The staining was performed in the dark at 4 °C for 20 min, and the percentage of mature DCs (CD11c^+^CD80^+^CD86^+^) was quantified by CytoFlex LX flow cytometer (Beckman Coulter, Inc., Brea, CA, USA). The data were analyzed using FlowJo V10 (Beckman Coulter, Inc., Brea, CA, USA) software.

For the analysis of T lymphocytes, lymphocytes underwent staining procedures. First, they were stained with Fixable Viability Dye eFluor^®^ 780. Subsequently, staining was carried out using FITC-conjugated anti-CD3, APC-conjugated anti-CD45, PE Cy7-conjugated anti-CD4, and PerCP-Cy5.5-conjugated anti-CD8. For the analysis of Tregs, a series of additional steps were performed. The cells were initially washed, followed by fixation and permeabilization. After these pre-treatments, they were stained with PE-conjugated anti-FoxP3 and BV421-conjugated anti-CD25. For the analysis of IFN-γ^+^ T lymphocytes, the cells were washed, fixed, permeabilized, and then stained with PE-conjugated anti-IFN-γ. Staining took place at 4 °C in permeabilization buffer for 20 min. After that, the cells were washed and analyzed with the CytoFlex LX flow cytometer. FlowJo V10 software was used for data analysis.

### 4.11. Candidate Peptide Synthesis and In Vitro Immunogenicity Assessment

The candidate peptides were synthesized by QYAOBIO (ChinaPeptides Co., Ltd., Shanghai, China) with a purity of >95%. The immunogenicity of the candidate peptides was examined through IFN-γ ELISPOT assays involving BALB/c mouse splenocytes. In brief, newly isolated splenocytes were placed in an incubation process with each of the candidate peptides at a concentration of 50 μg/mL, and during this process, 50 U/mL of IL-2 was included. Bone marrow-derived dendritic cells (BMDCs) were generated from six-week-old female BALB/c mice by adhering to a protocol that has been previously reported [79]. BMDCs were cultured in complete RPMI-1640 medium (Gibco, Billings, MT, USA) containing 10% FBS and 1% penicillin/streptomycin, GM-CSF (25 ng/mL), and IL-4 (10 ng/mL). On the sixth day, immature DCs were incubated overnight with each candidate peptide at a concentration of 50 μg/mL. The following day, splenocytes primed with the peptide (2 × 10⁵ cells per well) were co-cultivated with the peptide-loaded BMDCs (1 × 10^2^ cells per well) in 96-well ELISPOT plates that had been pre-coated with a capture antibody specific for IFN-γ. This co-culture was then maintained in an incubator set at 37 °C with a 5% CO_2_ atmosphere for 20 h. Once the incubation period was over, the cells were removed from the plates, and the plates were processed according to the guidelines provided by the manufacturer of the IFN-γ ELISPOT kit. Finally, the number of spots was counted using a spot reader system.

### 4.12. In Vitro BMDC Maturation Assay

BMDCs were generated from six-week-old female BALB/c mice using a previously reported protocol [79]. Briefly, BMDCs were cultured in RPMI-1640 medium supplemented with 1% penicillin/streptomycin, 10% FBS, GM-CSF (25 ng/mL), and IL-4 (10 ng/mL). On day 6, immature BMDCs were co-cultured overnight with the candidate immune peptides or the positive stimulant lipopolysaccharide (LPS, 1 μg/mL, 93572-42-0, Sigma–Aldrich, St. Louis, MO, USA). In order to evaluate the purity and maturity of the cultured BMDCs, the cells that were harvested on the seventh day were subjected to a staining process. The staining was performed at 4 °C in a dark environment using the fixable viability dye eFluor^®^ 780, PE-conjugated anti-mouse CD11c, PE Cy7-conjugated anti-mouse CD86, and APC-conjugated anti-mouse CD80, and this staining procedure lasted for 20 min. After the cells were washed, the quantification of the percentage of mature BMDCs (CD11c^+^CD80^+^CD86^+^) was carried out by utilizing the CytoFlex LX flow cytometer. Subsequently, the obtained data were analyzed with the assistance of the FlowJo V10 software.

### 4.13. Statistical Analysis

Statistical analysis was performed using GraphPad Prism (Version 8.3, San Diego, CA, USA). All data were presented as mean ± standard deviation (s.d.). For data with normal distribution and homogeneous variance, one-way ANOVA was used to evaluate differences between group means. When ANOVA showed statistical significance, pairwise comparisons were performed using the least significant difference (LSD) post hoc test. Student’s *t*-test was used when comparing 2 groups. Statistical significance was indicated by * *p* < 0.05, ** *p* < 0.01, *** *p* < 0.001, and **** *p* < 0.0001.

## 5. Conclusions

In conclusion, this study has demonstrated that ER-Cy-*po*NO_2_-mediated phototherapy enhances tumor immunogenicity by inducing ER stress, remodeling the immunopeptidome, and promoting the MHC-I presentation of immunogenic peptides, such as IF4G3_986–994_. These changes drive robust CD8^+^ T lymphocyte responses and inhibit tumor growth, highlighting the critical role of ER stress in enhancing antigenicity and anti-tumor immunity. Our findings provide new insights into the mechanisms by which ER-targeted phototherapy enhances ICD-based tumor immunogenicity, offering a strong foundation for the development of novel strategies in cancer vaccine research.

## Figures and Tables

**Figure 1 pharmaceuticals-18-00491-f001:**
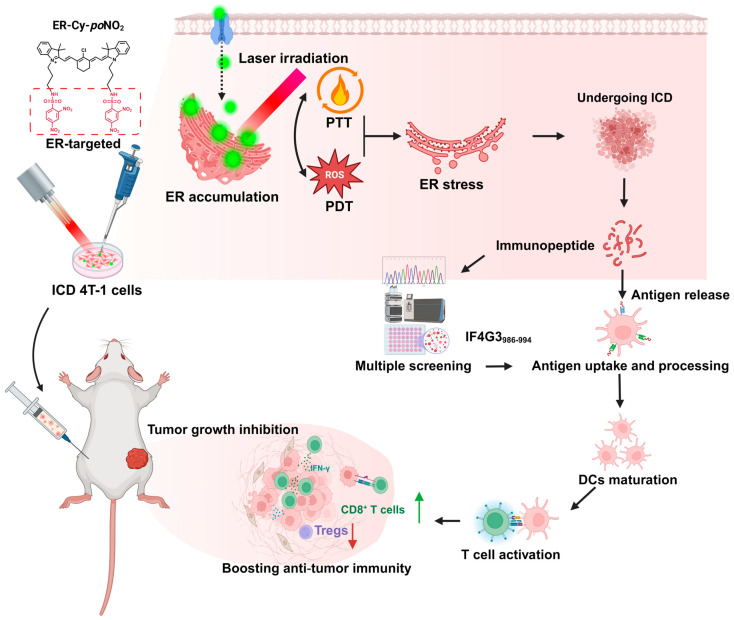
A schematic depiction of the underlying mechanism by which ER-targeted phototherapy remodels the tumor immunopeptidome to enhance immunogenic cell death and adaptive anti-tumor immunity. The photo-triggered immunotherapy is achieved by a tumor-ER dual-targeted photosensitizer (ER-Cy-*po*NO_2_) which can robustly induce the ER stress-driven ICD of tumor cells (ICD 4T-1 cells in this work). Consequently, substantial tumor-derived MHC-I binding peptides are produced. After multiple screenings of these peptides, IF4G3_986–994_ (QGPKTIEQI) is identified with an exceptional immunogenicity which enables it to effectively augment the anti-tumor immunity. The figure was created in BioRender. Gao, M. (2025) https://BioRender.com/p39u912, accessed on 19 February 2025.

**Figure 2 pharmaceuticals-18-00491-f002:**
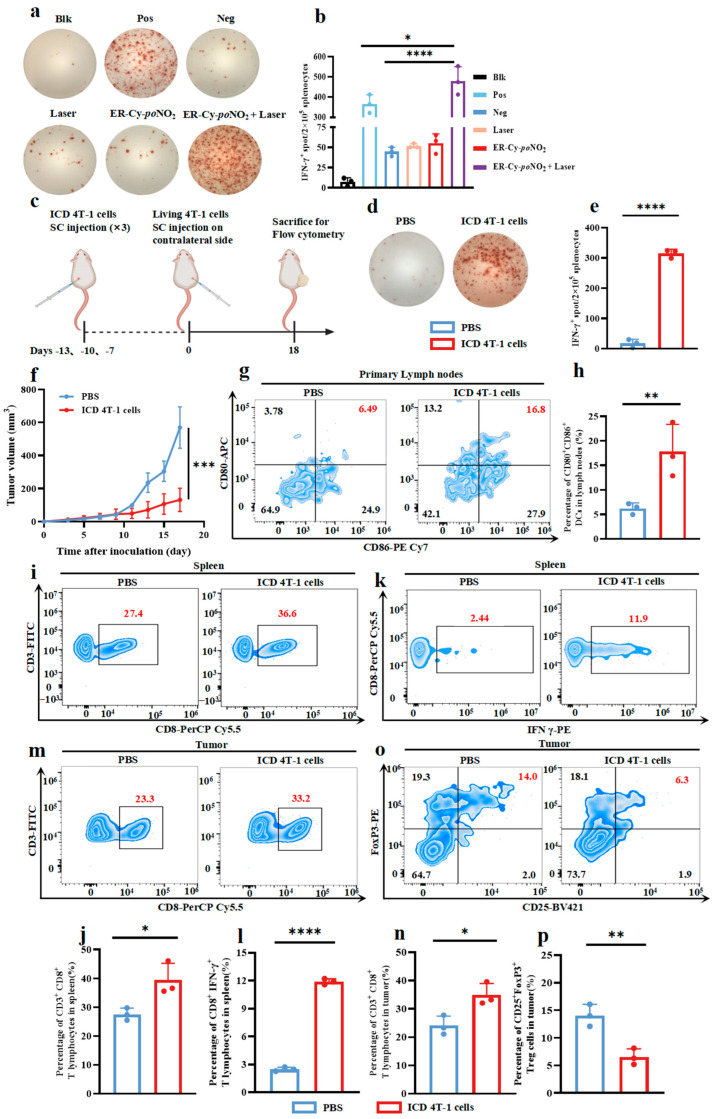
ER-targeted phototherapy enhanced the immunogenicity of 4T-1 tumor cells undergoing ICD. (**a**) Representative ELISPOT images showing IFN-γ spots after co-incubation of 4T-1 cells with splenocytes under different treatments. (**b**) Summary data for the assay in (**a**) showing the mean IFN-γ spot number per 2 × 10^5^ splenocytes ± s.d. in duplicate cultures. (*n* = 3, mean ± s.d.). (**c**) The workflow for the tumor vaccination experiment. (**d**) Representative ELISPOT images showing IFN-γ spots of splenocytes from mice inoculated with 4T-1 cells subjected to different treatments. (**e**) Summary data for the assay in (**d**) showing the mean IFN-γ spot number per 2 × 10^5^ splenocytes ± s.d. in duplicate cultures. (*n* = 3, mean ± s.d.). (**f**) Tumor volume curves during the observation period (*n* = 5, mean ± s.d.). (**g**) Representative images of flow cytometry analysis of DC maturation (CD80^+^CD86^+^) in primary lymph nodes. (**h**) Quantitative analysis of the proportion of CD80^+^CD86^+^ DCs in (g) (*n* = 3, mean ± s.d.). (**i**) Representative images of flow cytometry analysis of T lymphocytes (CD3^+^CD8^+^) in spleen. (**j**) Quantitative analysis of the proportion of CD3^+^CD8^+^ T lymphocytes in (**i**) (*n* = 3, mean ± s.d.). (**k**) Representative images of flow cytometry analysis of T lymphocytes (CD8^+^IFN-γ^+^) in spleen. (**l**) Quantitative analysis of the proportion of CD8^+^IFN-γ^+^ T lymphocytes in (k) (*n* = 3, mean ± s.d.). (**m**) Representative images of flow cytometry analysis of T lymphocytes (CD3^+^CD8^+^) in the tumor. (**n**) Quantitative analysis of the proportion of CD3^+^CD8^+^ T lymphocytes in (m) (*n* = 3, mean ± s.d.). (**o**) Representative images of flow cytometry analysis of Tregs (CD4^+^CD25^+^FoxP3^+^) in the tumor. (**p**) Quantitative analysis of the proportion of CD4^+^CD25^+^FoxP3^+^ Tregs in (**o**) (*n* = 3, mean ± s.d.). Statistical analyses were conducted using one-way ANOVA followed by Tukey’s test. Student’s *t*-test was used upon observing significant differences between groups. * *p* < 0.05, ** *p* < 0.01, *** *p* < 0.001, **** *p* < 0.0001.

**Figure 3 pharmaceuticals-18-00491-f003:**
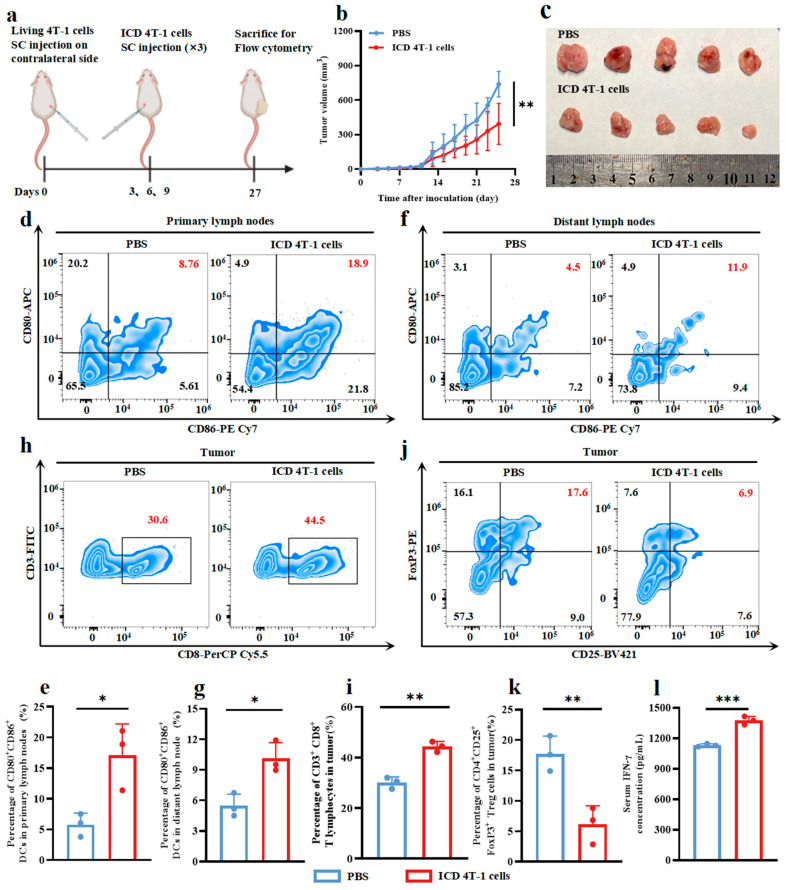
ICD tumor cells induced by ER-targeted phototherapy triggered a robust anti-tumor immunity in 4T-1-bearing mice. (**a**) The workflow for the tumor treatment experiment. (**b**) Tumor volume curves during the treatment period (*n* = 5, mean ± s.d.). (**c**) Images of tumors (*n* = 5). (**d**) Representative images of flow cytometry analysis of DC maturation (CD80^+^CD86^+^) in primary lymph nodes. (**e**) Quantitative analysis of the proportion of CD80^+^CD86^+^ DCs in (**d**) (*n* = 3, mean ± s.d.). (**f**) Representative images of flow cytometry analysis of DC maturation (CD80^+^CD86^+^) in distant lymph nodes. (**g**) Quantitative analysis of the proportion of CD80^+^CD86^+^ DCs in (**f**) (*n* = 3, mean ± s.d.). (**h**) Representative images of flow cytometry analysis of T lymphocytes (CD3^+^CD8^+^) in the tumor. (**i**) Quantitative analysis of the proportion of CD3^+^CD8^+^ T lymphocytes in (h) (*n* = 3, mean ± s.d.). (**j**) Representative images of flow cytometry analysis of Tregs (CD4^+^CD25^+^FoxP3^+^) in the tumor. (**k**) Quantitative analysis of the proportion of CD4^+^CD25^+^FoxP3^+^ Tregs in (**j**) (*n* = 3, mean ± s.d.). (**l**) Quantitative analysis of the concentration of IFN-γ in serum (*n* = 3, mean ± s.d.). Student’s *t*-test was used upon observing significant differences between groups. * *p* < 0.05, ** *p* < 0.01, and *** *p* < 0.001.

**Figure 4 pharmaceuticals-18-00491-f004:**
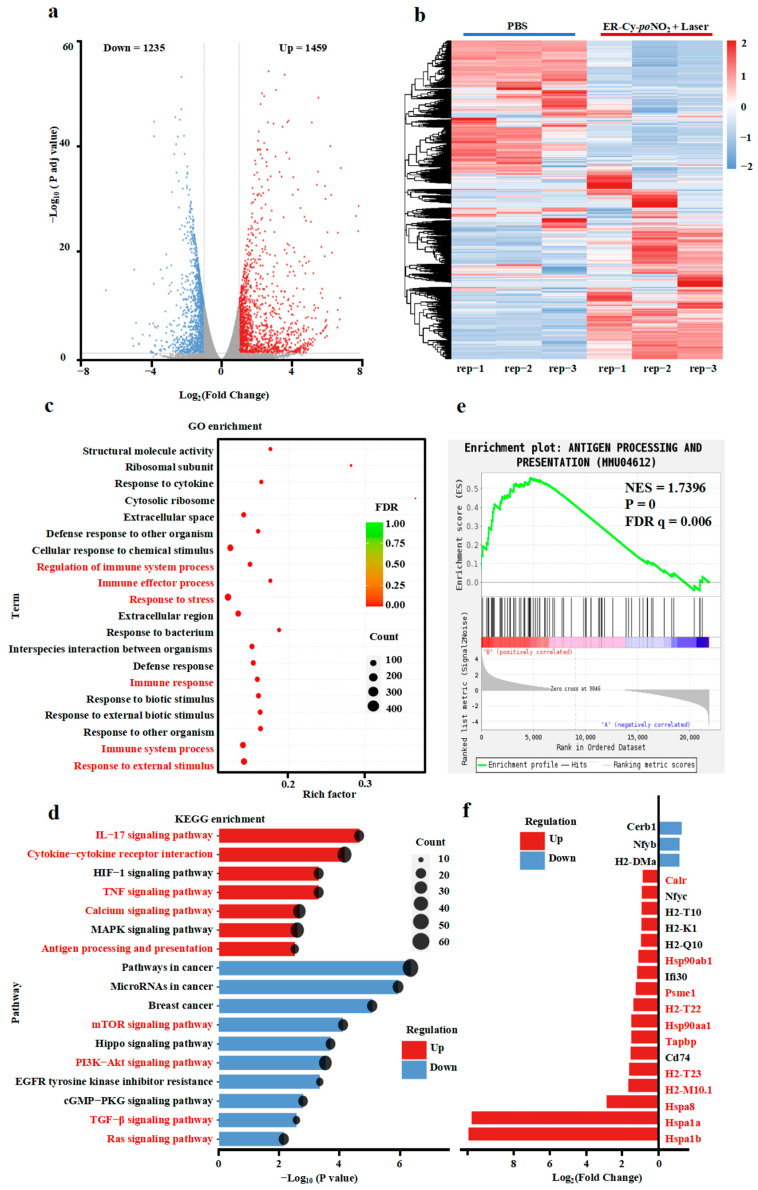
ER-targeted phototherapy upregulated antigen processing and immune signaling pathways in 4T-1 cells. (**a**) Volcano plots demonstrate the upregulated and downregulated DEGs in the ER-Cy-*po*NO_2_ + Laser group compared with the PBS group, with red indicating high expression and blue indicating low expression levels (log_2_|Fold change| > 1, *p* adj. value < 0.05). (**b**) The heatmap demonstrates the upregulated and downregulated DEGs, with red indicating high expression and blue indicating low expression levels. (**c**) The GO enrichment analysis of upregulated DEGs in the ER-Cy-*po*NO_2_ + Laser group compared with the PBS group. (**d**) The KEGG enrichment analysis of DEGs in the ER-Cy-*po*NO_2_ + Laser group compared with the PBS group. (**e**) The GSEA enrichment analysis of antigen processing and presentation (MMU04612) in the ER-Cy-*po*NO_2_ + Laser group compared with the PBS group. (**f**) The expression differences in key genes in pathway antigen processing and presentation (MMU04612).

**Figure 5 pharmaceuticals-18-00491-f005:**
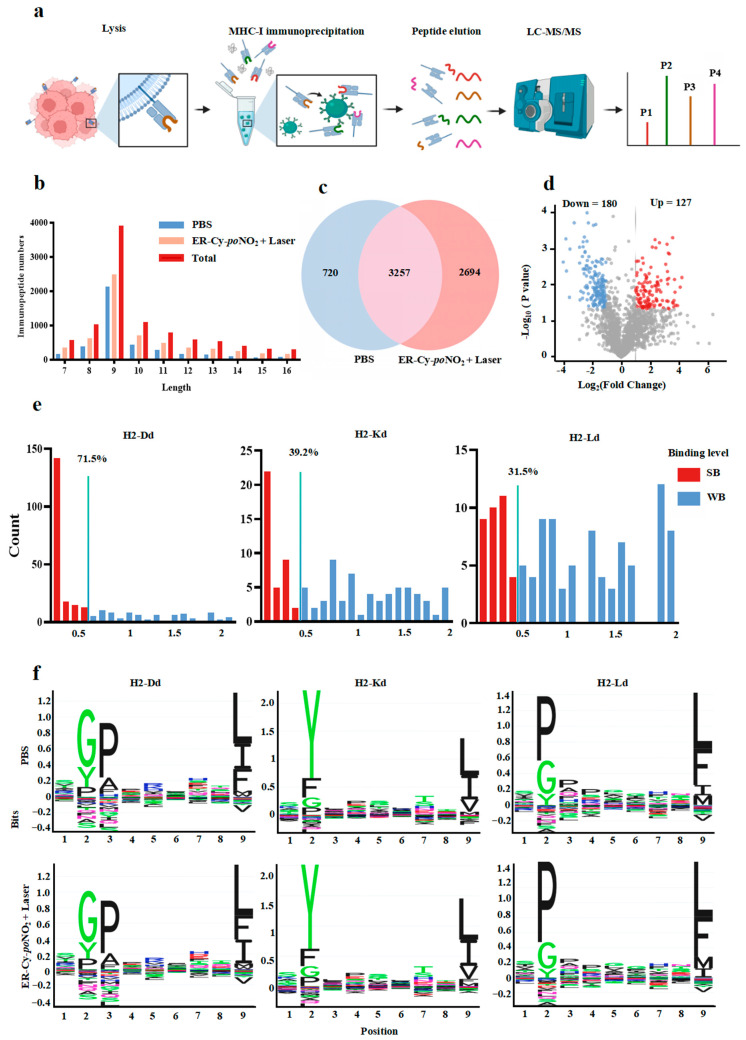
ER-targeted phototherapy remodeled the immunopeptidome of 4T-1 cells generating upregulated high-affinity MHC-I ligands. (**a**) The workflow for identifying the 4T-1 cell-derived MIP. (**b**) Length distributions of MIP derived from the PBS group, ER-Cy-*po*NO_2_ + Laser group, and total (7–16 amino acids). (**c**) The overlap of the MIP between the PBS group and ER-Cy-*po*NO_2_ + Laser group. (**d**) Volcano plots demonstrate the upregulated and downregulated differentially expressed MIP in the ER-Cy-*po*NO_2_ + Laser group compared with the PBS group among shared immunopeptides in (**c**), with red indicating high expression and blue indicating low expression levels (log_2_|Fold change| > 1, P value < 0.05). (**e**) The predicted affinities of peptides by NetMHCpan-4.1, SB: strong binder (% Rank < 0.5); WB: weak binder (%Rank:0.5–2). (**f**) The binding motifs of the 9-mer peptides identified in the 4T-1 cell-derived MIP with different treatments. The x-axis represents the residue position within the 9-mer peptide sequence. The y-axis represents the information content, with the size of each amino acid symbol proportional to its frequency.

**Figure 6 pharmaceuticals-18-00491-f006:**
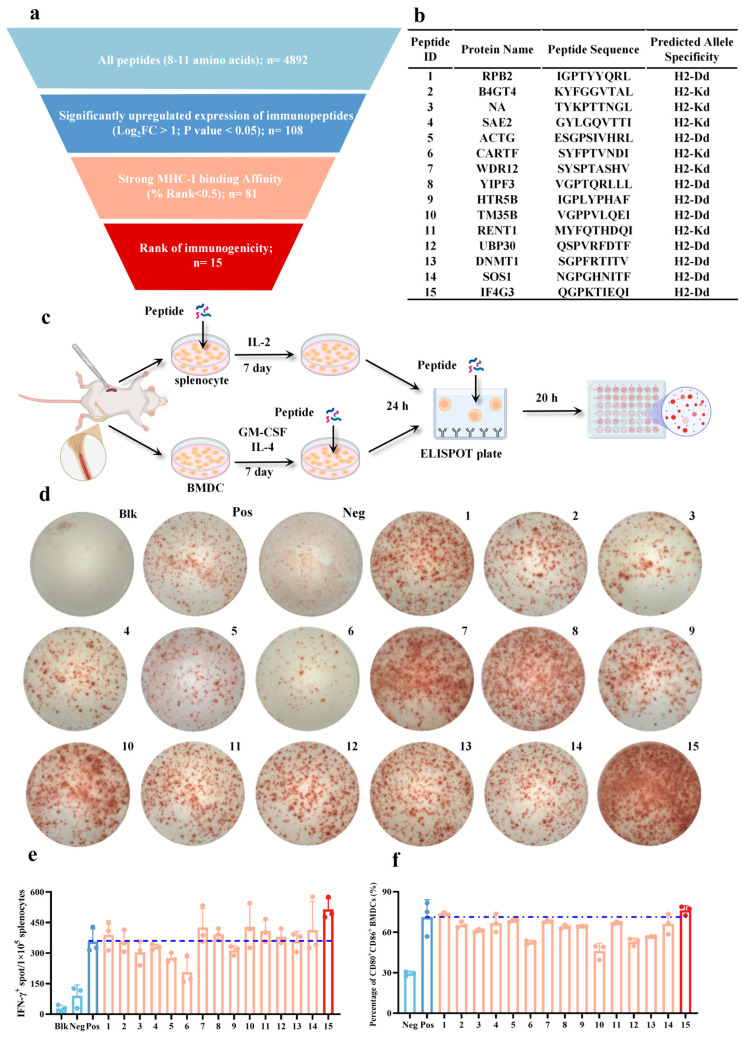
The IF4G3_986–994_ peptide demonstrated a pronounced immunogenicity advantage over the other tested candidates. (**a**) The illustration depicts the pipeline for immunopeptides’ selection for the immunogenicity assessment. (**b**) The illustration depicts the amino acid sequences, source proteins, and haplotypes of the 15 synthesized candidate immunopeptides. (**c**) The workflow for the recall IFN-γ ELISPOT assay. (**d**) Representative ELISPOT images showing IFN-γ produced by the indicated peptide-primed BALB/c mice splenocytes restimulated with BMDCs alone (BMDCs + no peptide) or the indicated peptide-pulsed BMDCs (BMDCs + indicated peptide). The blk group refers to the blank control (i.e., wells containing splenocytes and culture medium without any peptide stimulation). The pos group refers to the positive control (i.e., wells containing splenocytes and PMA plus ionomycin). The neg group refers to the negative control (i.e., wells containing splenocytes and BMDCs without peptide stimulation). (**e**) The summary data for the assay in (**d**) showing the mean IFN-γ spots number per 1 × 10^5^ splenocytes ± s.d. in duplicate cultures (*n* = 3, mean ± s.d.). (**f**) Quantitative analysis of the proportion of CD80^+^CD86^+^ BMDCs in Appendix A (*n* = 3, mean ± s.d.).

**Figure 7 pharmaceuticals-18-00491-f007:**
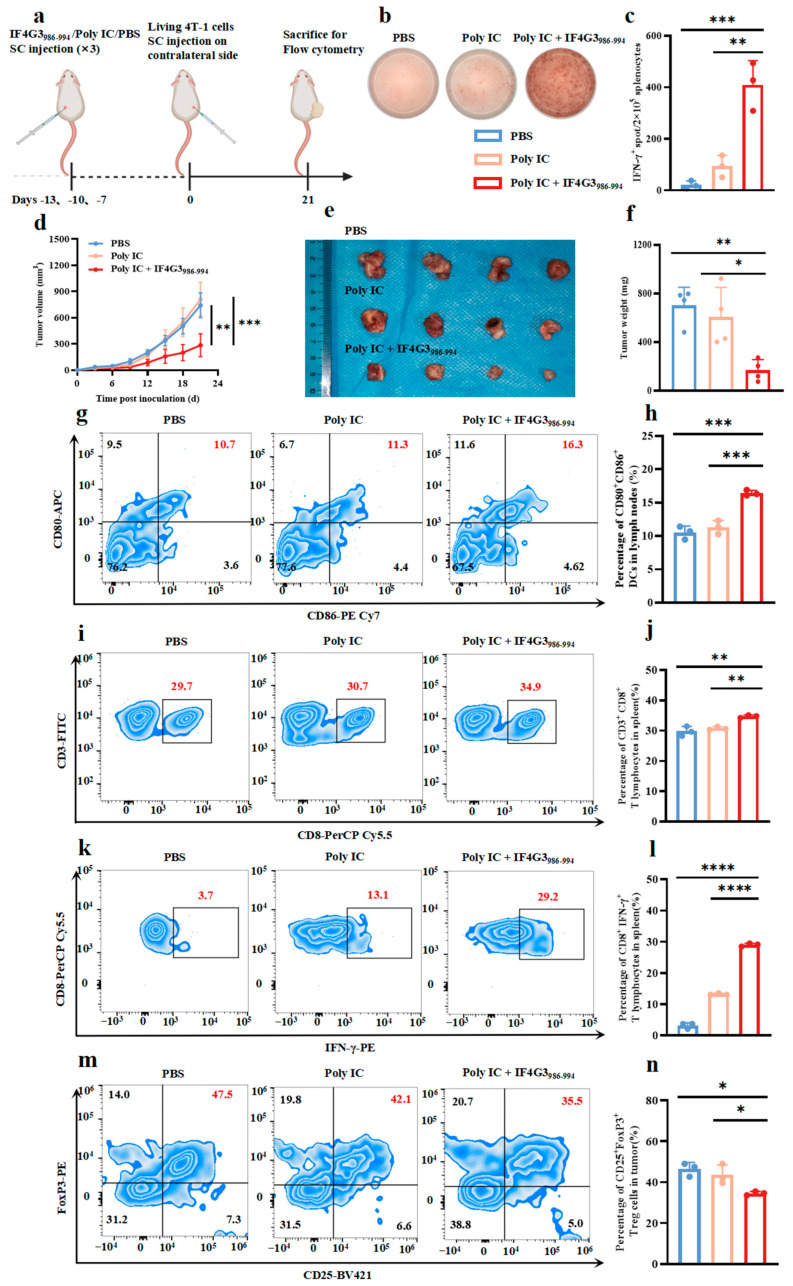
The highly immunogenic peptide IF4G3_986–994_ could effectively activate the anti-tumor immune response and subsequently inhibit tumor growth in mice. (**a**) The workflow for the IF4G3_986–994_ in vivo tumor vaccine experiment. (**b**) Representative ELISPOT images showing IFN-γ spots of splenocytes from mice subjected to different treatments. (**c**) Summary data for the assay in (**b**) showing the mean IFN-γ spot number per 2 × 10^5^ splenocytes ± s.d. in duplicate cultures. (*n* = 3, mean ± s.d.). (**d**) Tumor volume curves during the treatment period (*n* = 4, mean ± s.d.). (**e**) Images of tumors (*n* = 4). (**f**) Tumor weights on day 21 after various treatments (*n* = 4, mean ± s.d.). (**g**) Representative images of flow cytometry analysis of DCs’ maturation (CD80^+^CD86^+^) in lymph nodes. (**h**) Quantitative analysis of the proportion of CD80^+^CD86^+^ DCs from (g) (*n* = 3, mean ± s.d.). (**i**) Representative images of flow cytometry analysis of T lymphocytes (CD3^+^CD8^+^) in the spleen. (**j**) Quantitative analysis of the proportion of CD3^+^CD8^+^ T lymphocytes from (**i**) (*n* = 3, mean ± s.d.). (**k**) Representative images of flow cytometry analysis of T lymphocytes (CD8^+^IFN-γ^+^) in the spleen. (**l**) Quantitative analysis of the proportion of CD8^+^IFN-γ^+^ T lymphocytes from (k) (*n* = 3, mean ± s.d.). (**m**) Representative images of flow cytometry analysis of Tregs (CD4^+^CD25^+^FoxP3^+^) in the tumor. (**n**) Quantitative analysis of the proportion of CD4^+^CD25^+^FoxP3^+^ Tregs from (**m**) (*n* = 3, mean ± s.d.). Statistical analyses were conducted using one-way ANOVA followed by Tukey’s test. * *p* < 0.05, ** *p* < 0.01, *** *p* < 0.001, and **** *p* < 0.0001.

## Data Availability

Data are contained within the article.

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
